

# Genome-wide identification, expression pattern and interacting protein analysis of INDETERMINATE DOMAIN (IDD) gene family in *Phalaenopsis equestris*

Ting Zhang[1,2], Xin Yu[1], Da Liu[1], Deyan Zhu[1,2] and Qingping Yi[1,2]

[1] College of Bioengineering, Jingchu University of Technology, Jingmen, Hubei, China
[2] Hubei Engineering Research Center for Specialty Flowers Biological Breeding, Jingchu University of Technology, Jingmen, Hubei, China

## ABSTRACT

The plant-specific *INDETERMINATE DOMAIN* (*IDD*) gene family is important for plant growth and development. However, a comprehensive analysis of the *IDD* family in orchids is limited. Based on the genome data of *Phalaenopsis equestris*, the *IDD* gene family was identified and analyzed by bioinformatics methods in this study. Ten putative *P. equestris IDD* genes (*PeIDD*s) were characterized and phylogenetically classified into two groups according to their full amino acid sequences. Protein motifs analysis revealed that overall structures of PeIDDs in the same group were relatively conserved. Its promoter regions harbored a large number of responsive elements, including light responsive, abiotic stress responsive elements, and plant hormone *cis*-acting elements. The transcript level of *PeIDD* genes under cold and drought conditions, and by exogenous auxin (NAA) and abscisic acid (ABA) treatments further confirmed that most *PeIDD*s responded to various conditions and might play essential roles under abiotic stresses and hormone responses. In addition, distinct expression profiles in different tissues/organs suggested that *PeIDD*s might be involved in various development processes. Furthermore, the prediction of protein-protein interactions (PPIs) revealed some PeIDDs (PeIDD3 or PeIDD5) might function via cooperating with chromatin remodeling factors. The results of this study provided a reference for further understanding the function of *PeIDD*s.

## INTRODUCTION

Spatio-temporal specific expression of genes is the basis of cell differentiation in multicellular organisms, in which transcription factors (TFs) play dominant roles in controlling gene expression by recognizing and binding special *cis*-elements. The *INDETERMINATE DOMAIN* (*IDD*) genes encode a plant-specific family of zinc finger TFs, which is characterized by a conserved ID domain that consists of two C2H2 and two C2HC fingers (*Colasanti et al., 2006*). *IDD* genes have been identified in many plants

Corresponding authors
Deyan Zhu, zhudeyan@jcut.edu.cn
Qingping Yi, yiqingping@jcut.edu.cn

and are widely reported to involve in plant growth and development, and adaptation to environment.

The identification and function analysis of *IDD* genes originated from crop and model plant *Arabidopsis thaliana*. Sixteen, fifteen, and twenty-two *IDD* genes have been identified in Arabidopsis, rice, and maize, respectively (*Coelho et al., 2018*; *Feng et al., 2023*; *Zhang et al., 2020*). These *IDD* genes are involved in different developmental processes and stress response. In root development, AtIDD3/MAGPIE (MAG) and AtIDD10/JACKDAW (JKD) were reported to regulate tissue boundary formation by cooperating with SHR (SHORT-ROOT) and SCR (SCARECROW) (*Long et al., 2015*; *Ogasawara et al., 2011*). *OsIDD10* was involved in the seminal root development of rice seedling *via* mediating N-linked metabolic responses (*Xuan et al., 2013*). During seed formation process, *AtIDD1/ENHYDROUS* (*ENY*) regulated light and hormonal signal pathway and eventually promoted seed germination (*Feurtado et al., 2011*). Duplicated *IDD* genes (*ZmIDDveg9* and *ZmIDD9*) encoded NKD (naked endosperm) proteins that were required for maize aleurone cell fate and cell differentiation (*Yi et al., 2015*; *Gontarek et al., 2016*). To the leaf and shoot development, *AtIDD4/IMPERIAL EAGLE* and other four *AtIDD* s (*AtIDD5*, *AtIDD10*, *AtIDD11*, and *AtIDD14*), as downstream genes of REV (REVOLUTA), involved in the ad/abaxial regulatory network (*Reinhart et al., 2013*). *OsIDD14/Loose Plant Architecture1* (*LPA1*), the ortholog gene of *AtIDD15*, had distinct functions from *AtIDD15* (*Cui et al., 2013*; *Liu et al., 2016*). As a part of the complicated regulatory network, *OsIDD14* affected the sedimentation rate of amyloplasts and shoot gravitropism of rice (*Wu et al., 2013*). GA homeostasis and sugar metabolism are associated with flowering. AtIDD8 promoted flowering by activating the expression of two sucrose synthesis genes (*SUS1* and *SUS4*) (*Seo et al., 2011b*). *AtIDD2* (*GAI-ASSOCIATED FACTOR1*, *GAF1*) encodes a DELLA-Interacting protein. Overexpression of *GAF1* resulted in flowering earlier, while *gaf1* flowered slightly later than WT (wild type) under short-day conditions. The GA-related phenotype of *gaf1* suggested GA responsiveness was affected (*Fukazawa et al., 2014*). The variations in sucrose and starch levels revealed the cause of an extreme late-flowering phenotype of *zmid1* (*Coneva et al., 2012*). In addition, *OsID1*, a rice ortholog gene of *ZmID1*, acted as a flowering promoter by regulating flowering-related gene expression (*Matsubara et al., 2008*). For stress response, *AtIDD14* generated two splice variants (AtIDD14α and AtIDD14β). The functional AtIDD14α had DNA binding activity and could promote starch degradation. AtIDD14β form was cold-induced produced and bound with AtIDD14α to repress AtIDD14α function so as to help plants to tolerate low temperature (*Seo et al., 2011a*). Recent research suggested AtIDD14 also regulated drought tolerance by interacting with bZIP-type ABFs/AREBs (*Liu et al., 2022*). AtIDD4 was identified as a negative regulator of salt stress, and overexpression of *AtIDD4* resulted in hypersensitive to salt-stress (*Rawat et al., 2023*). ROC1 (regulator of CBF1), an IDD protein of rice, was verified as a chilling tolerance regulator based on the cold sensitive phenotype and lower level of *CBF1* transcripts in *roc1* mutant (*Dou et al., 2016*). OsIDD12, OsIDD13, and OsIDD14 could form a transcription complex to activate the expression of *MDPK* (*Malectin Domain Protein Kinase*) to enhance resistance of *ShB* (*Sheath blight*) (*Cui et al., 2022*).

With the deepening of the research, *IDD* genes gained more and more attention and had been explored in ornamental plants. In a comparative genomic analysis of the *IDD* gene family in five Rosaceae species, 16 *IDD* genes were determined in Chinese white pear (*Pyrus bretschneideri*) and most of *PbIDD* s have a high transcription level in productive organs (*Su et al., 2019*). The comprehensive analysis of *IDD* genes in apple identified 20 putative *IDD* genes and these *IDD* genes responsed to various circumstances (*Fan et al., 2017*). Peach (*Prunus persica*) has *14* IDD genes, and PpIDD4, PpIDD12, and PpIDD13 can interact with DELLA1, a vital factor in GA signaling pathway, to participate in GA feedback regulation (*Jiang et al., 2022*).

*Phalaenopsis equestris*, known for its butterfly-shaped flowers and brilliant colors, is a widely cultivated flowers and of great economic importance. Despite extensive studies of *IDD* genes were under way, little is known about the features of *PeIDD* s. Due to the genome sequence of *P. equestris* has emerged recently (*Cai et al., 2015*), the opportunity arises to conduct the comprehensive study of the *PeIDD* family. In this study, 10 *PeIDD* s were identified from *P. equestris.* Phylogenetic relationship, gene structure, protein structure and gene expression profile of 10 *PeIDD* s were carried out. More importantly, we predicted the interaction proteins of PeIDDs. The results will provide a certain theoretical basis for further research on th functions of *PeIDD*s.

## MATERIALS AND METHODS

### Identification of *IDD*s in *P. equestris* and phylogenetic analysis

The *P.equestris* genome sequence and annotation file were download from the National Center for Biotechnology Information (NCBI, https://www.ncbi.nlm.nih.gov/genome/?term=txid78828[orgn]&shouldredirect=false). AtIDD and OsIDD proteins were downloaded from The Arabidopsis Information Resource (TAIR) database (https://www.arabidopsis.org) and The Rice Genome Annotation Project Database and Resource (http://rice.uga.edu/index.shtml) (Table S1), respectively. To identifiy potential IDDs in *P. equestris*, 16 AtIDD and 15 OsIDD proteins were used as query sequences in a BLAST search with default parameters in TBtools v1.120 software (*Chen et al., 2020*). In addition, all candidate PeIDD proteins were verified according to the method of previous report (*Jiang et al., 2022*) to confirm IDD domains. Incomplete and redundant protein sequences were discarded manually. Subsequently, physicochemical properties (PIs, MWs) were predicted through the ExPASy website (https://www.expasy.org/).

Multiple sequence alignment of IDD proteins from *Oryza sativa*, *A. thaliana* and *P. equestris* was done using ClustalX 2.0 software (*Larkin et al., 2007*). The sequence alignment results were used to construct the neighbor-joining (NJ) tree by MEGA 7.0 software (*Kumar, Stecher & Tamura, 2016*), with the following parameters: Poisson correction, complete deletion, and bootstrap (1,000 replicates).

### Gene structures analysis, conserved motifs prediction, and *cis*-element analysis of the promoters

The gene structures of *PeIDD*s were determined by DNA and cDNA sequence alignment. The potentially conserved motifs of PeIDD proteins, including ID domain, were determined

by the online Multiple Expectation for Motif Elicitation (MEME, http://meme-suite.org), (*Bailey et al., 2009*), using the following parameter settings: the maximum number of motifs, 10; the minimum width of motifs, 5; the maximum width of motifs, 20; and other default parameters. The promoter sequences of *PeIDD*s were harvested from NCBI website and its *cis*-acting elements were predicted using the PlantCARE database (http://bioinformatics.psb.ugent.be/webtools/plantcare/html/) (*Lescot et al., 2002*). All gene structures, conserved motifs, and *cis*-elements were visualized by running the IBS version 1.0.2 software (*Liu et al., 2015*).

## Protein interaction prediction

The online STRING platform (https://string-db.org) (*Szklarczyk et al., 2017*) was used to predict the protein-protein network between PeIDDs and other proteins, using *Dendrobium catenatum* as reference organism, with the parameter: predicted interacting protein score above 0.4.

## Plant materials, RNA extraction and qRT-PCR

Fifteen *P. equestris* was collected from Lufa Orchid (Taiwan, China) and placed in the greenhouse of Jingchu University of Technology. Roots, leaves, flowers, and floral stalks were sampled and immediately stored at −80 °C. For plant hormone and stress treatment, the plants were exposed to 0.2 M ABA, $10^{-6}$ M NAA, 20% PEG6000, and 4 °C. Total RNA from different tissues was isolated by TransZol (ET101-01-V2; TransGen Biotech Co., Ltd., Beijing, China) reagent, and detected by NANODROP 1,000 spectrophotometer to determine the extraction quality, then reversed transcribed into cDNA with HiScript II Q RT SuperMix for qPCR (+gDNA wiper) (R223-01; Vazyme, Hangzhou, China) following the manufacturer's instructions. The cDNA was diluted to 100 ng/μL and used as a template for RT-qPCR. Specific primers for 10 *PeIDD* genes with amplified size ranging from 170 to 250 bp were designed using online programs (https://sg.idtdna.com/scitools/Applications/RealTimePCR/) (Table S2). The RT-qPCR was performed on the QuantStudio 6 Flexreal-time PCR instrument (Applied Biosystems, Foster City, CA, USA). *PeActin* was used as an internal reference (F: GCTGAGGGAGGCAAGGATAGAT; R: GCACCCAGCAGCATGAAGATC) to standardize gene expression levels, and each cDNA was subjected to three biological replicates. The PCR mixture (10 μL) included 5 μL *Taq* Pro Universal SYBR qPCR Master Mix (Q712-02; Vazyme, Hangzhou, China), 10 mM of each primer, 100 ng cDNA template, and nuclease-free water. The PCR program was performed as follows: 95 °C for 10s; 40 cycles of 95 °C for 5 s, 60 °C for 40 s. Relative expression values were calculated using the $2^{-\Delta\Delta Ct}$ method (*Livak & Schmittgen, 2001*) and visualized by GraphPad Prism 8.0.2.

# RESULTS

## Genome-wide identification and characterization of *PeIDD* gene family

Using 16 AtIDD and 15 OsIDD protein sequences as the queries, 12 PeIDD proteins were initially identified by BLAST searches. Examined by ID domains, ten proteins were finally

confirmed as PeIDD family members. According to the order distribution on the scaffolds (Fig. S1), these genes werer named PeIDD1-PeIDD10. The analysis of these PeIDD protein sequences (Table S3) revealed that the physicochemical properties of each member were different (Table S4). The amino acid lengths of the PeIDDs ranged from 197 aa (PeIDD3) to 480 aa (PeIDD1). The molecular weight (MW) ranged from 22,098.33 Da (PeIDD3) to 50,799.25 Da (PeIDD1). The isoelectric point (pI) of PeIDDs ranged from 8.71 (PeIDD10) to 9.56 (PeIDD3). The instability index spanned from 37.79 (PeIDD8) to 89.71 (PeIDD3).

## Phylogenetic analysis of PeIDDs

To better understand the phylogenetic relationship of IDDs in *A. thaliana* (16 members), *O. sativa* (15 members) and *P. equestris* (10 members), a neighbor-joining (NJ) tree of 41 IDD proteins was constructed based on their full length amino acid sequences. All IDD proteins were clearly divided into two groups (I and II) (Fig. 1). Group I had a large number of member, with 13 AtIDDs, 12 OsIDDs, and 8 PeIDDs, accounting for 80.5% of the total IDD proteins. While group II only harbored 8 IDD proteins (three AtIDDs, three OsIDDs, and two PeIDDs). On the basis of the bootstrap values and topological structure, group I can be divided into four subgroups. Interesting, subgroup I-2 were solely composed of AtIDDs and OsIDDs. In other subgroups of group I (subgroup I-1, I-3, and I-4), most PeIDDs were clustered with OsIDDs firstly, suggesting that PeIDDs might have a close relationship to OsIDDs.

## Gene structure and motif composition of *PeIDD* gene family

To gain more insight into the potential relationship about gene structure-function and protein structure-function, gene structure and motif prediction of PeIDDs were determined. For the exon-intron structures, most *PeIDD*s displayed three exons, whereas *PeIDD1* and *PeIDD7* had four and two exons, respectively (Fig. 2A). The length of exons was similar, while the intron lengths varied greatly (Fig. 2A). In particular, *PeIDD6* and *PeIDD7* had exceptionally long introns, which was similar to the *KNOX* gene structure of orchids and might be unique to Orchidaceae (*Zhang et al., 2022*). In *P.equestris* genome, half of *PeIDDs* (*PeIDD3*, *PeIDD4*, *PeIDD6*, *PeIDD8* and *PeIDD10*) had no 5′-UTR, 3′-UTR, and both UTRs (5′-UTR and 3′-UTR) (Fig. 2A). This suggested that they were likely the UTR-less *IDD* genes in *P. equestris*.

Subsequently, MEME (http://meme-suite.org) was used to predict the putative motifs shared among PeIDD proteins. In total, 8 types of motifs were observed (Fig. 2B, Fig. S2). Motif 2, 3, 4, and 5 were widespread at the N-terminus of all PeIDDs. Motif 2 encoded a nuclear localization signal (NLS), indicating that all PeIDDs are located in the nucleus. Motif 3, 4, and 5 constituted the conserved ID domain, which was the unique structure of IDDs. Motif 1 was specific to the PeIDDs (PeIDD2, PeIDD4, and PeIDD9) of subgroup I-1 (Fig. 1). Motif 6, following the ID domain, only existed in PeIDD1, PeIDD2, PeIDD4, and PeIDD9. Motif 7 ("MSATALLQKAA" domain) and motif 8 ("TRDFLG" domain) that were verified to act in protein-protein interactions (*Colasanti et al., 2006*) were found in the C terminus of most PeIDDs. Because motifs are associated with protein interactions or binding to target genes, differences in motif composition suggest that these PeIDDs may function in different mechanisms.

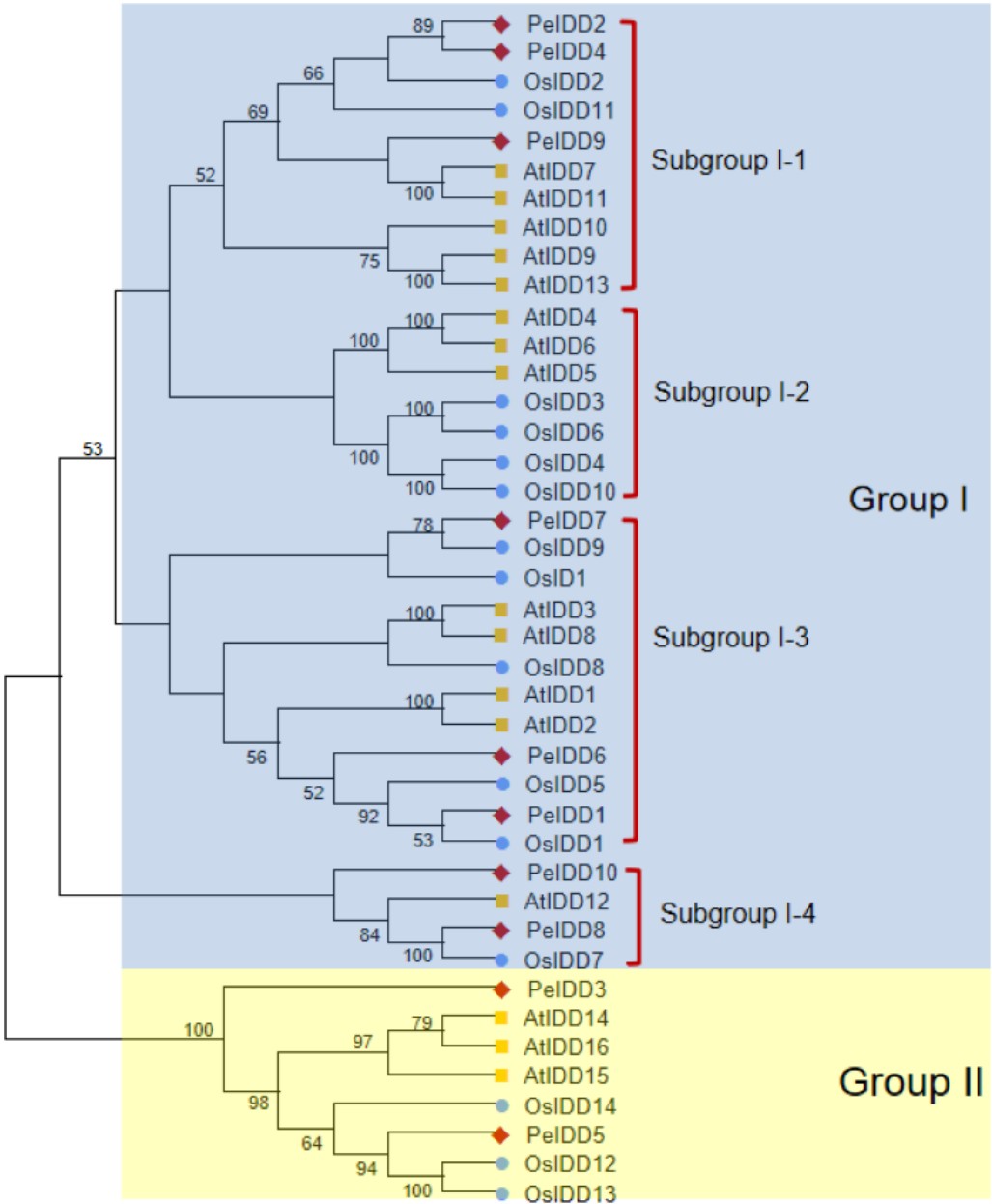

**Figure 1** **Phylogeny of IDD homologs in *Phalaenopsis equestris* (Pe,red diamond), *Oryza sativa* (Os, blue circles), and *Arabidopsis thaliana* (At, yellow squares).** Cluster X and MEGA 7.0 were used to align the protein sequences and generate the neighbor-joining (NJ) tree, respectively. Bootstrap values (1,000 replicates) more than 50% were shown on branches.

### *Cis*-regulatory elements analysis of *PeIDD* genes

In order to a space between the two words in the promoter regions of *PeIDD*s, 2,000 bp sequences upstream of transcription start site of *PeIDD*s were retrieved as the promoters

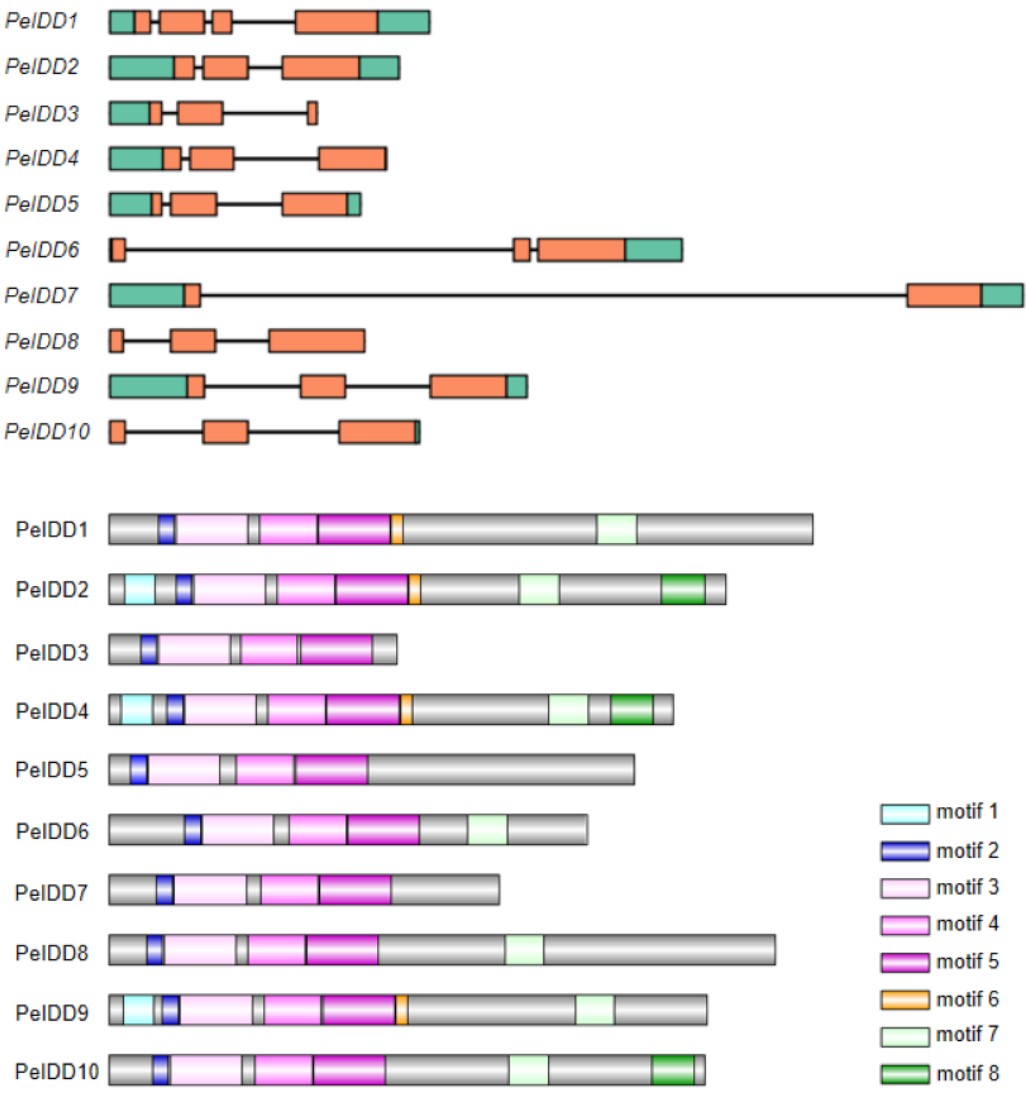

**Figure 2  Gene structures and conserved motifs in *PeIDDs*.** (A) Gene structures were visualized using IBS1.0.2. UTRs and exons were represented by green and red boxes, respectively. (B) Conserved motifs were identified using the online MEME program and represented by different colored boxes. Motif sequences were shown in Fig. S2.

(Table S5), and then submitted them to the PlantCARE database. A large number of *cis*-elements were obtained and could be classified into three types: growth and development-related, phytohormone responsive, and abiotic stress responsive (Table S6). Growth and development-related elements consisted of LRE (light responsive element), SSE (seed-specific regulation element), MEE (meristem expression element), PEE (palisade mesophyll cells expression element), and EEE (endosperm expression element). Hormone responsive elements included ABA, GA, Auxin, SA, and MeJA. Abiotic stresses responsive elements were low-temperature responsive element and drought-related element (Fig. 3).

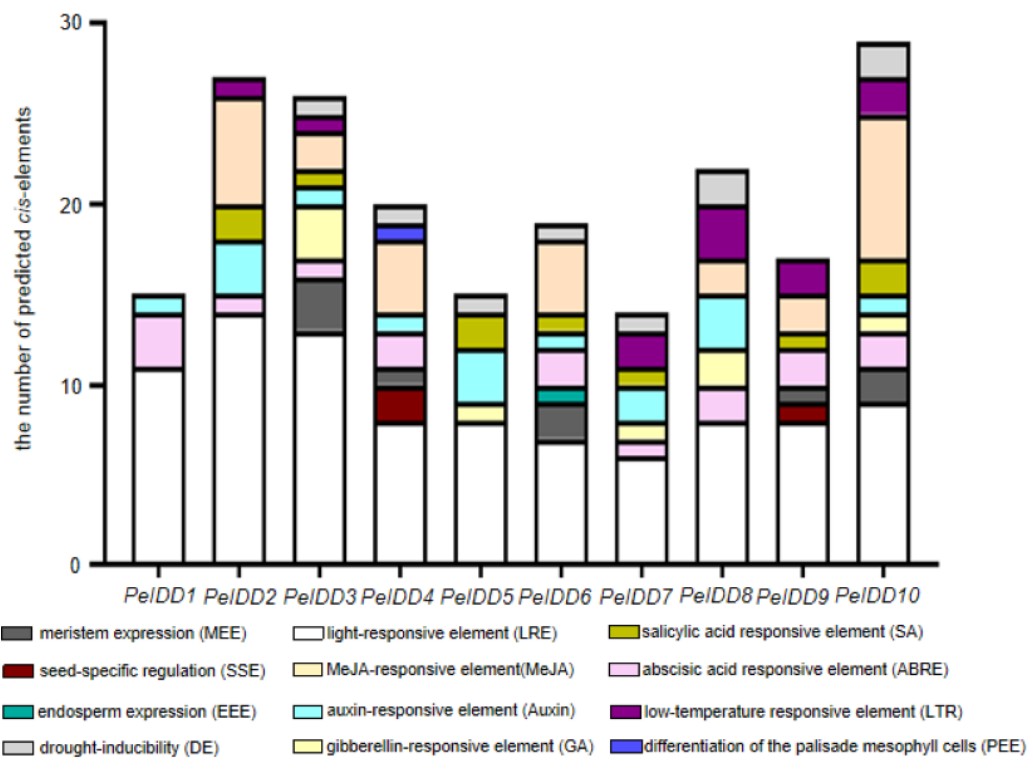

**Figure 3** **The types and numbers of predicted cis-elements in the promoter of the *PeIDD* genes.** Details information of predicted *cis*-elements in *PeIDD* genes were listed in Table S6 . Various *cis*-elements were shown by different colored boxes.

Among these growth and development-related *cis*-elements, LREs were the most frequently occurring responsive elements, suggesting that *PeIDD*s might play important roles in *P. equestris* light morphogenesis. Plant hormones are also essential to plant growth and development. ABA-responsive elements (ABRE) and Auxin-responsive elements (TGA-element, AuxRR-core) were observed in nine *PeIDD*s. SA responsive elements (TCA-element) and MeJA responsive elements (TGACG-motif, CGTCA-motif) existed in seven *PeIDD*s. GA responsive elements (P-box, TATC-box, GRAE-motif) were found in five *PeIDD*s. The widespread distribution of hormone response elements in the promoter indicated that *PeIDD*s might be widely involved in hormone signal pathway.

The adaptability to the environment is a focus in plant breeding. Two *cis*-elements related to abiotic stress, namely, drought response element (LTR) and low temperature response element (DE), appeared in the promoter of *PeIDD*s. DE existed in seven *PeIDD*s, and LTR was present in six *PeIDD*s. These results suggested that *PeIDD*s might play a vital role in stress adaptation.

### Expression patterns of *PeIDD* genes in different tissues

The tissue-specific expression patterns of the *PeIDD* gene family were investigated in four tissues/organs, including root, flower, leaf, and floral stalk. The results revealed that the *PeIDD*s exhibited different expression level in these tissues. Seven *PeIDD*s (*PeIDD2*,

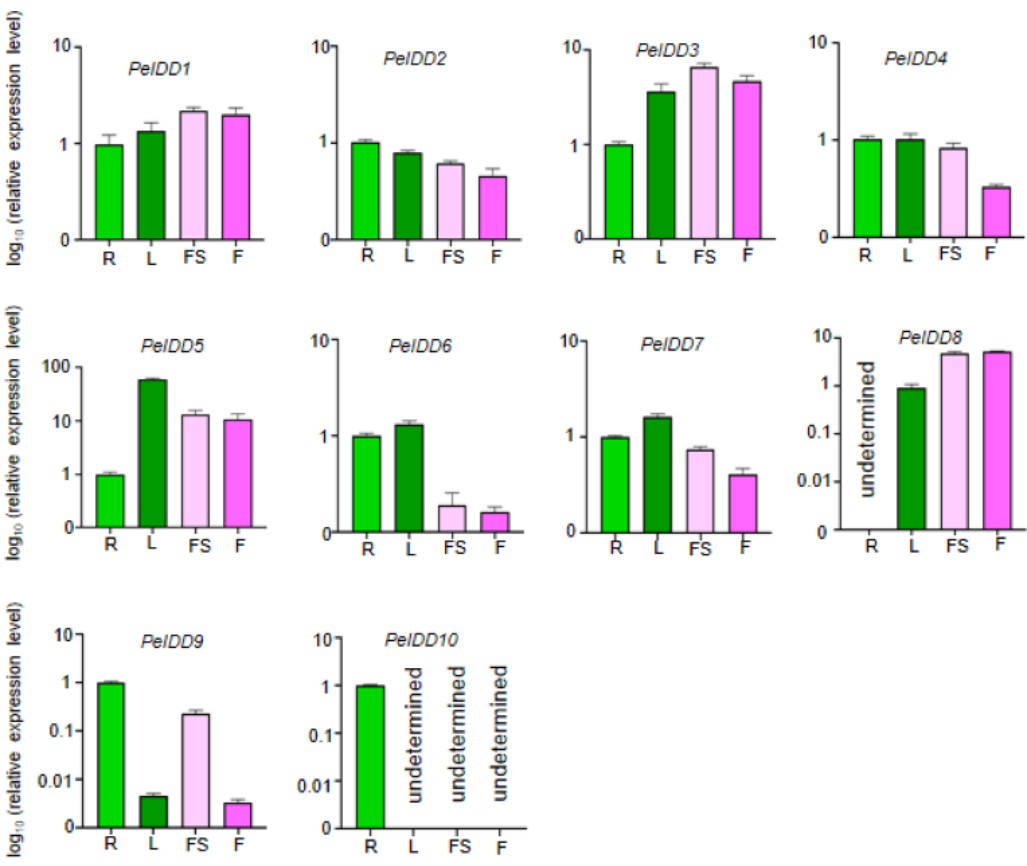

**Figure 4** The expression profiles of *PeIDD* genes in roots (R), leaves (L), floral stalks (FS), and flowers (F). Three independent experiments were performed and the means were represented by error bars. *PeIDD*s transcript levels were normalized to *PeActin*.

*PeIDD4*, *PeIDD5*, *PeIDD6*, *PeIDD7*, *PeIDD9*, and *PeIDD10*) showed high transcripts in vegetative tissues (root and leaf), but relatively low expression levels in the reproductive tissue (flower and floral stalk) (Fig. 4). Among them, *PeIDD5* was predominantly expressed in leaf (Fig. 4), hinting *PeIDD5* might participate in leaf development. *PeIDD9* and *PeIDD10* were highly expressed in the root, especially *PeIDD10* was expressed specifically in the root and barely detected in other tissues (Fig. 4), indicating *PeIDD9* and *PeIDD10* might regulate root development. The transcript levels of *PeIDD1*, *PeIDD3*, and *PeIDD8* displayed higher expression in floral stalk and flower (Fig. 4), suggesting they might be involved in flower development.

## Response of *PeIDD* genes to plant hormones and abiotic stress

Because of lots of hormone and stress responsive elements were found in the promoter of *PeIDD*s, the expression patterns of *PeIDD*s under hormones (ABA and NAA) and stresses (drought and cold) during the different time courses (0 h, 1 h, 5 h, and 10 h) were analyzed by RT-qPCR. Under 0.2 M ABA treatment, seven *PeIDD*s (except *PeIDD3*, *PeIDD7*, and *PeIDD10*) expression levels reached their peaks after 1 h or 5 h. *PeIDD7* was

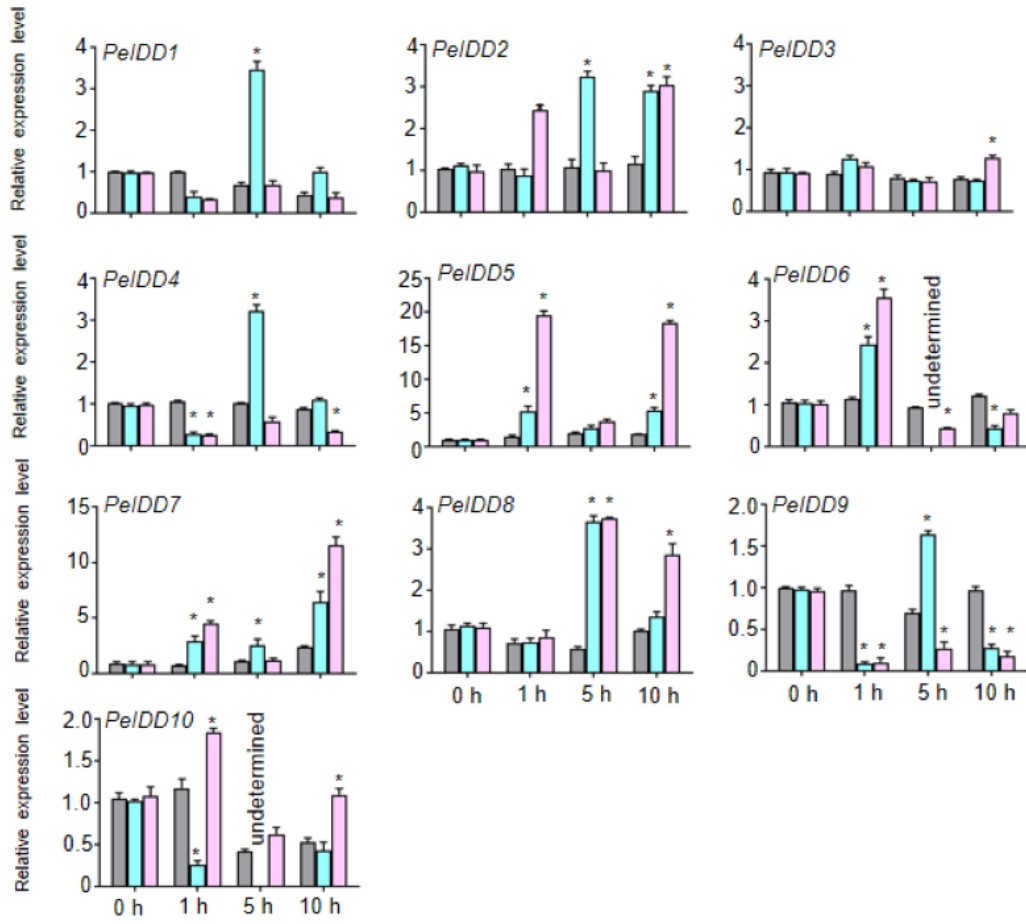

**Figure 5** **The expression patterns of *PeIDD*s under ABA and NAA treatment.** *PeIDD* transcript levels were determined by RT-qPCR under different plant hormone. Grey, blue and pink bars represented untreated, ABA treated, and NAA treated groups, respectively. The PCR signals were normalized with those of *PeActin* transcripts. The SDs (standard deviations) of the means of three independent biological replicates were denoted by error bars. Statistically significant differences between untreated and treatment groups were analyzed by an independent Student's *t*-tests and shown by asterisks (* $P < 0.05$).

induced at 1 h and peaked after 10 h, while *PeIDD3* showed no significant changes and the expression of *PeIDD10* was inhibited (Fig. 5). After NAA treatment 1 h, *PeIDD1*, *PeIDD4* and *PeIDD9* transcripts had a significant decline compared with the untreated group, while the expression levels of *PeIDD2*, *PeIDD5*, *PeIDD6*, and *PeIDD7* showed an over two-fold increase. *PeIDD8* was obviously upregulated after NAA treatment 5 h (Fig. 5).

Regardless of whether *PeIDD*s had drought responsive elements or not (Fig. 3, Table S4), all *PeIDD*s responded to drought at different time points. Nine *PeIDD*s showed at least two-fold increase of transcripts, while *PeIDD4* exhibited down-regulation of transcript level. The results revealed that *PeIDD*s might be widely involved in drought response. Under cold treatment, these results were related to the number of LTRs in the promoter of *PeIDD*s (Fig. 3, Table S4). For example, four *PeIDD*s (*PeIDD7*, *PeIDD8*, *PeIDD9*, and *PeIDD10*) contained two or three LTRs and showed obvious induced expression (Fig. 6),

1234

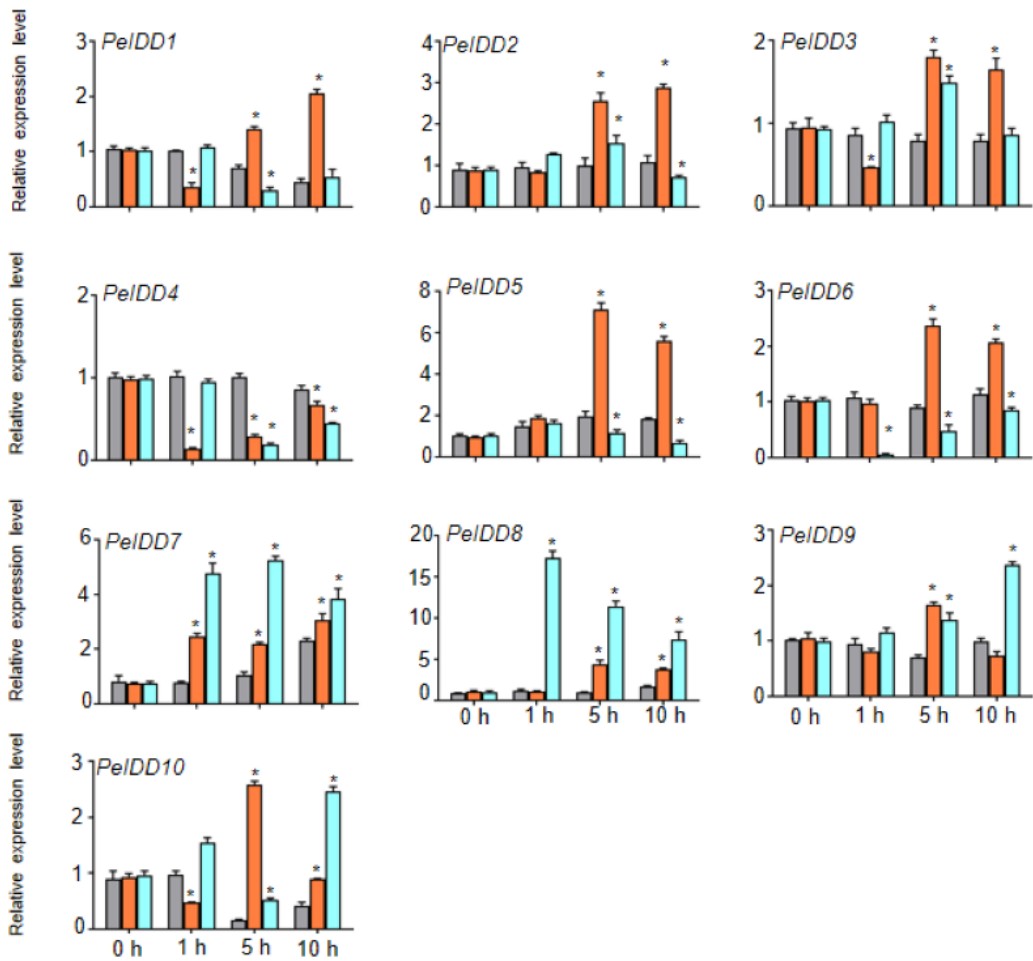

**Figure 6  The expression patterns _PeIDD_s under drought and cold stress.** _PeIDD_ transcript levels were determined by RT-qPCR under different abiotic stresses. Grey, red and blue bars represented untreated, drougt, and cold stress groups, respectively. The PCR signals were normalized with those of PeActin transcripts. The SDs (standard deviations) of the means of three independent biological replicates were denoted by error bars. Statistically significant differences between untreated and treatment groups were analyzed by an independent Student's $t$-tests and shown by asterisks (*$P < 0.05$).

among which the expression of _PeIDD_ 8 increased more than 8-fold after 1 h of treatment. _PeIDD2_ and _PeIDD3_ had one LTR and exhibited a slight increase in expression. However, _PeIDD1_, _PeIDD4_, _PeIDD5_, and _PeIDD6_ had no LTR elements, their expression level was down-regulated after cold stress (Fig. 6), which suggested that they might be regulated indirectly or had other unknown cold responsive elements.

## Prediction of protein interaction network of _PeIDD_ proteins

Although the function of PeIDDs has not been reported, understanding the protein–protein interaction (PPI) network may help us understand their regulatory mechanism. Based on the prediction results from the STRING website (Table S5), the interaction network was constructed by Cytoscape software. The results showed PeIDDs could interact with 21 proteins, including TFs (5), chromatin remodeling factors (4), enzymes (2), and other

proteins (10) (Fig. 7, Table S5). Among TFs, ERF021 and TINY are ethylene responsive factors (*Xie et al., 2019*). PeIDD2 and PeIDD10 could interact with ERF021 and PeIDD4 with TINY, suggesting that the three PeIDDs might be integrated into the ethylene response pathway. WRKY family is known to involve in biotic and abiotic stress response (*Wani et al., 2021*; *Khoso et al., 2022*). WRKY24 was predicted to interact with PeIDD4 and PeIDD9, indicating PeIDD4 and PeIDD9 might participate in stress adaptation. PTI5 is a pathogenesis-related transcriptional activator (*Wang et al., 2021*). PeIDD5 might be involved in plant immune response by interacting with PTI5. Among chromatin remodeling factors, SWI/SNF (SWITCH/SUCROSE NONFERMENTING) was reported to regulate chromatin structure (*Bieluszewski et al., 2023*). PeIDD3 and PeIDD5 were predicted to act with SWI3 subunit, indicating PeIDD3 and PeIDD5 might regulate transcript levels of their target genes by recruiting chromatin remodeling factors. The interaction of PeIDD8 to SMC3 (structural maintenance of chromosomes protein 3) indicated that PeIDD8 might help to stabilize chromosome structure. ATP-dependent helicase BRM is one of the enzymes that PeIDDs bound. Its interaction protein PeIDD5 might help it locate to specific chromatin regions. Dehydration-responsive element-binding protein (DREB) and the SHR-SCR (SHORTROOT-SCARECROW) complex stood out among the other proteins. PeIDD2, PeIDD4, PeIDD9, and PeIDD10 could interact with DREB, indicating that they could participate in drought responses. SHR-SCR complex was reported to involve in root development (*Shaar-Moshe & Brady, 2023*), suggesting that PeIDD6 and PeIDD7 might be in a common regulatory pathway with it. These results fully demonstrated the function diversity of PeIDDs and the complexity of the regulatory pathways.

## DISCUSSION

*P. equestris* is an ornamental plant and known for its elegant appearance and extended longevity. A comprehensive understanding of *P. equestris* IDD gene family, and making use of them, will help to enhance the growth and ornamental value of *P. equestris*. Here, a total of ten *PeIDD* genes were identified in the *P. equestris* genome, their expression profiles in different tissues, under hormone and stress treatment, and interaction proteins information were determined.

According to protein sequence similarity, all PeIDDs identified in this study were classified into two groups (group I and II) (Fig. 1), which was consistent with the classification of IDD proteins in rice (*Zhang et al., 2020*), *Phyllostachys edulis* (*Guo et al., 2022*), and Arabidopsis (*Coelho et al., 2018*). These IDD proteins in different groups/subgroups might have been functionally diverged and involved in different biological processes. In subgroup I-2, AtIDD4, AtIDD6 and OsIDD10 were related to root development (*Moreno-Risueno et al., 2015*; *Yoshida et al., 2014*; *Xuan et al., 2013*). In subgroup I-3, OsID1 and OsIDD1 are flowering regulator (*Wu et al., 2008*; *Deng et al., 2017*). Among PeIDD proteins in the same subgroup, PeIDD1 clustered with OsIDD1 firstly and PeIDD7 with OsID1 (Fig. 1). Combined higher expression level of *PeIDD1* in flower (Fig. 4), we speculated that *PeIDD1* might be involved in flowering of *P. equestris*. As flowering factors controlling the phase transition from vegetative to reproductive phase,

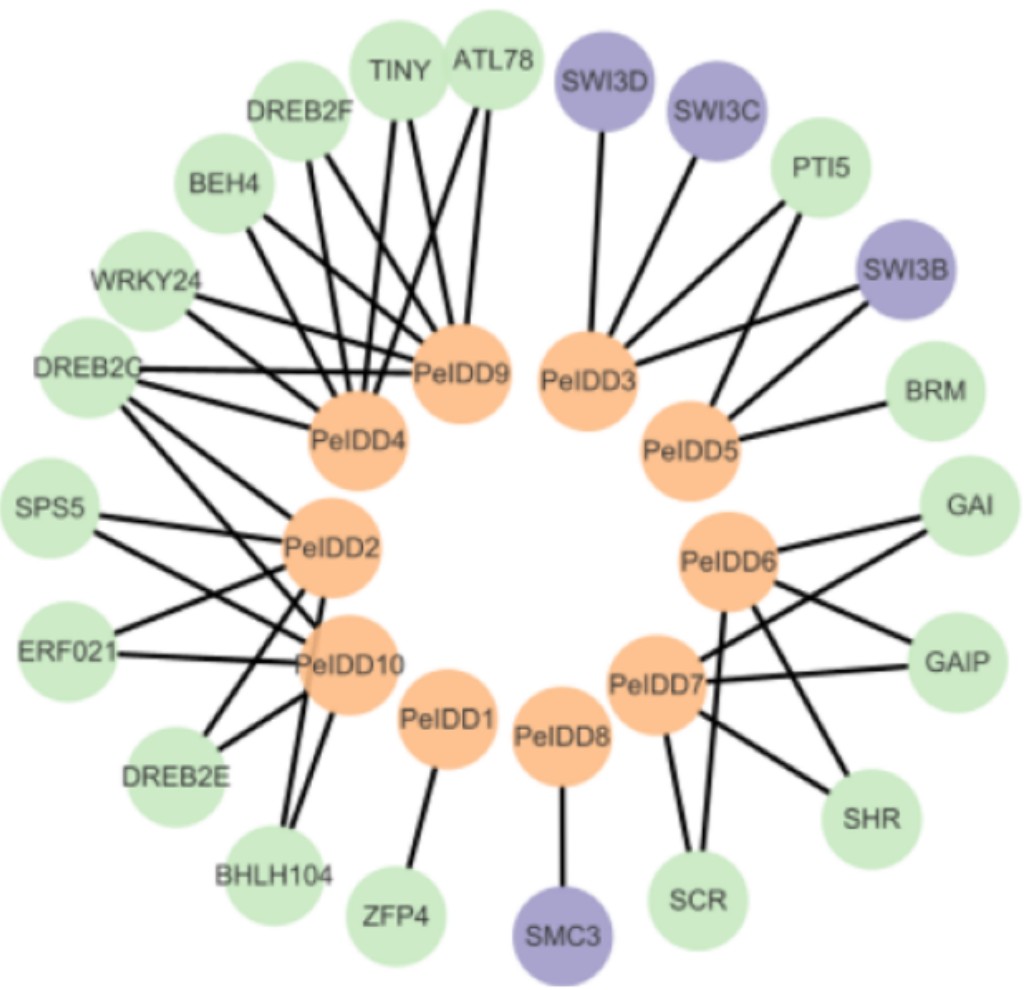

**Figure 7  Protein–protein interaction network of *PeIDD*s.** Orange, purple, and green circles represented *PeIDD*s, epigenetic factors (SWI3) and structural maintenance of chromosomes (SMC3), and other proteins, respectively. The information of interacting proteins was shown in Table S7.

OsID1 and OsIDD1 proteins differ in motif composition. OsIDD1 had motif 7 at the C-terminus, OsID1 did not (*Zhang et al., 2020*). This indicated that there might be differences in regulating rice flowering mechanism between OsID1 and OsIDD1. PeIDD1 also had motif 7 (Fig. 2), suggesting PeIDD1 might have similar mechanism of promoting flowering to OsIDD1. Taken together, PeIDD1 might be a potential target gene for regulating flower development in *P. equestris*, which needs further experiments to verify its function.

The *cis*-elements in the promoter determined the response of genes to environmental cues. The promoter of *PeIDD* genes contained a variety of *cis*-acting elements (Fig. 4, Table S4), which suggested that *PeIDD*s might function in various physiological processes and response to different signals. Hence, the expression level of all *PeIDD* genes to plant hormones and abiotic stresses were investigated by RT-qPCR (Figs. 5 and 6) and are mainly

consistent with the *cis*-elements. Interesting, although not all *PeIDD* genes have drought-responsive elements, all *PeIDD*s responded to drought stress, whether their expression levels increased or decreased (Fig. 6). This indicated *PeIDD*s directly or indirectly involved in drought response, consistent with previous reports that *IDD*s might be related to the formation of Kranz ring in C4 plants and important in drought resistance (*Coelho et al., 2018*). Among *PeIDD* genes, *PeIDD8* had the most cold-responsive elements (Table S4) and showed a rapid response to cold stress, with a sharp transcript level increase (Fig. 6). Low temperature seriously inhibits the growth of *P. equestris.* Based on the response of *PeIDD8* to cold stress, *PeIDD8* might act as an effective regulators to improve cold stress resistance/tolerance in *P. equestris.*

PPI prediction have been thought as an important content in gene family analysis, because it can help us to understand the molecular mechanism of protein function. In *Brassica napus*, *Wang et al. (2018)* predicted 38 proteins interacted with BnWOXs, including peptides, TFs, and other proteins. In rice, *Zhang et al. (2020)* predicted histone modifiers could interact with OsIDDs which indicated that OsIDDs might regulate the expression of downstream target genes through changing the chromatin structure. Interestingly, PPI analysis revealed that some PeIDDs (PeIDD3 and PeIDD5) could cooperate with chromatin remodeling factors (Fig. 7). Both chromatin remodeling factors and histone modifiers are epigenetic modifiers with the ability to alter chromatin structure. Consequently, interaction with epigenetic modifiers to regulate the expression of downstream genes might be a common regulation mode of IDDs.

## CONCLUSIONS

In this study, ten *PeIDD*s were characterized and classified into two groups based on protein sequence and conserved motifs at the C-terminal. Expression profiles of *PeIDD*s under plant hormones and abiotic stresses suggested that *PeIDD*s might widely participate in hormone/abiotic stress signaling pathway. Importantly, PPIs analysis revealed some PeIDDs might interact with chromatin remodeling factors to modulate target genes expression. Taken together, our studies provided a theoretical basis for further analysis of the molecular mechanism of *PeIDD*s in *P. equestris.*

### Funding
This research was supported by the Opening Project of Hubei Engineering Research Center for Specialty Flowers Biological Breeding (No. 2022ZD007) and the Hubei Provincial Central Government-Led Local Science and Technology Development special project (No. 2022BGE262). The funders had no role in study design, data collection and analysis, decision to publish, or preparation of the manuscript.

### Grant Disclosures
The following grant information was disclosed by the authors:

Opening Project of Hubei Engineering Research Center for Specialty Flowers Biological Breeding: 2022ZD007.
Hubei Provincial Central Government-Led Local Science and Technology Development Special Project: 2022BGE262.

## Competing Interests

The authors declare there are no competing interests.

## Author Contributions

- Ting Zhang conceived and designed the experiments, performed the experiments, analyzed the data, prepared figures and/or tables, authored or reviewed drafts of the article, and approved the final draft.
- Xin Yu performed the experiments, prepared figures and/or tables, and approved the final draft.
- Da Liu performed the experiments, prepared figures and/or tables, and approved the final draft.
- Deyan Zhu conceived and designed the experiments, authored or reviewed drafts of the article, collecting different tissue samples, and approved the final draft.
- Qingping Yi conceived and designed the experiments, analyzed the data, authored or reviewed drafts of the article, collecting different tissue samples, and approved the final draft.

## Data Availability

The raw measurements are available in the Supplementary Files.

## Supplemental Information

Supplemental information for this article can be found online at http://dx.doi.org/10.7717/peerj.18073#supplemental-information.

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
