# Peer review of "Genome-wide identification, expression pattern and interacting protein analysis of INDETERMINATE DOMAIN (IDD) gene family in Phalaenopsis equestris"

_PeerJ, doi:10.7717/peerj.18073_

## Round 0.1 · original submission · Major Revisions

Thank you for your submission to PeerJ.

There are comments from reviewers. Please revise the manuscript as per their comments.

Reviewer 1 ·

Basic reporting

The authors identified the IDD gene family in Phalaenopsis equestris. They analyzed the gene structure and expression patterns. Some questions need be revised.
1. The language must be polished.
2. Which species is selected as reference in protein interaction analysis.
3. What standard used to classify groups (I and II) in phylogenetic analysis.
4. How to estimate relative expression? In M&M section, authors hold that 2-Ct method used to estimate relative expression. If yes, the relative expression of control group is 1 (20). However, in figures 4-6, all relative expression levels of control group is not 1. Please clarify it.

Experimental design

no comment'

Validity of the findings

no comment'

·

Basic reporting

This review provides a thorough evaluation of the manuscript, identifying both strengths and areas for improvement. By addressing these comments, the authors can enhance the clarity, robustness, and impact of their work.

Experimental design

Major Comments
• The study would benefit significantly from functional validation experiments. Consider including knockout or overexpression studies for key PeIDD genes to validate their roles in development and stress responses. These experiments would provide direct evidence supporting the bioinformatic predictions and expression data.
• The discussion section should delve deeper into the potential functional implications of the identified PeIDD genes. Specifically, discuss how these genes might influence orchid biology and horticulture practices. Highlight any potential applications in breeding or genetic modification for improved stress tolerance and growth characteristics.
• While the phylogenetic analysis compares PeIDDs with IDDs from Arabidopsis and rice, further comparative functional analysis with these model plants could strengthen the conclusions. Discuss how the similarities and differences observed might reflect functional divergence or conservation among species.

Validity of the findings

Minor Comments
• Ensure that the manuscript is thoroughly proofread to correct minor grammatical errors and improve sentence structure for better readability. For instance, revising complex sentences to be more concise and clear can enhance comprehension.
• Some figure legends could benefit from additional detail to make them more informative. For example, explicitly stating what each color or symbol represents in the phylogenetic tree and motif diagrams would improve clarity.
• Provide a clearer description of how raw data can be accessed and used by other researchers. This transparency will facilitate replication and further studies based on your findings.

Reviewer 3 ·

Basic reporting

In this paper, the IDD gene family of Phalaenopsis equestris was identified and analyzed, which is of great significance for IDD gene mining, but certain modifications should be made before acceptance, as follows:
1 It is suggested to add collinearity analysis
2 line43-44,This sentence is not appropriate, the author did not carry out a comprehensive analysis of the gene family, such as the lack of collinearity analysis, kaks analysis, etc
3 It is suggested to redraw the Fig. S1
4 The author should simplify some of the language in the article, for example, line 313-315, this sentence is meaningless
5 The English in the whole text should be improved as necessary, for example,line 36, The word genes should be followed by the word were

Experimental design

In this paper, the IDD gene family of Phalaenopsis equestris was identified and analyzed, which is of great significance for IDD gene mining, but certain modifications should be made before acceptance, as follows:
1 It is suggested to add collinearity analysis
2 line43-44,This sentence is not appropriate, the author did not carry out a comprehensive analysis of the gene family, such as the lack of collinearity analysis, kaks analysis, etc
3 It is suggested to redraw the Fig. S1
4 The author should simplify some of the language in the article, for example, line 313-315, this sentence is meaningless
5 The English in the whole text should be improved as necessary, for example,line 36, The word genes should be followed by the word were

Validity of the findings

In this paper, the IDD gene family of Phalaenopsis equestris was identified and analyzed, which is of great significance for IDD gene mining, but certain modifications should be made before acceptance, as follows:
1 It is suggested to add collinearity analysis
2 line43-44,This sentence is not appropriate, the author did not carry out a comprehensive analysis of the gene family, such as the lack of collinearity analysis, kaks analysis, etc
3 It is suggested to redraw the Fig. S1
4 The author should simplify some of the language in the article, for example, line 313-315, this sentence is meaningless
5 The English in the whole text should be improved as necessary, for example,line 36, The word genes should be followed by the word were

Additional comments

In this paper, the IDD gene family of Phalaenopsis equestris was identified and analyzed, which is of great significance for IDD gene mining, but certain modifications should be made before acceptance, as follows:
1 It is suggested to add collinearity analysis
2 line43-44,This sentence is not appropriate, the author did not carry out a comprehensive analysis of the gene family, such as the lack of collinearity analysis, kaks analysis, etc
3 It is suggested to redraw the Fig. S1
4 The author should simplify some of the language in the article, for example, line 313-315, this sentence is meaningless
5 The English in the whole text should be improved as necessary, for example,line 36, The word genes should be followed by the word were

---

## Round 0.2 · accepted · Accept

Thank you for your submission to PeerJ.
Congratulations!

·

Basic reporting

The article meets the PeerJ criteria and should be accepted as is.

Experimental design

Nill

Validity of the findings

Nill

Additional comments

Nill

Reviewer 3 ·

Basic reporting

The author made careful revisions to the article and agreed to publish it

Experimental design

The author made careful revisions to the article and agreed to publish it

Validity of the findings

The author made careful revisions to the article and agreed to publish it

Additional comments

The author made careful revisions to the article and agreed to publish it